# Modifications in Gene Expression in the Process of Osteoblastic Differentiation of Multipotent Bone Marrow-Derived Human Mesenchymal Stem Cells Induced by a Novel Osteoinductive Porous Medical-Grade 3D-Printed Poly(ε-caprolactone)/β-tricalcium Phosphate Composite

**DOI:** 10.3390/ijms222011216

**Published:** 2021-10-18

**Authors:** Ivan López-González, Camilo Zamora-Ledezma, María Isabel Sanchez-Lorencio, Elena Tristante Barrenechea, José Antonio Gabaldón-Hernández, Luis Meseguer-Olmo

**Affiliations:** 1Tissue Regeneration and Repair Group, Orthobiology, Biomaterials and Tissue Engineering, Campus de los Jerónimos 135, UCAM-Universidad Católica de Murcia, Guadalupe, 30107 Murcia, Spain; czamora9@ucam.edu; 2Biomedical Research Institute of Murcia (IMIB-Arrixaca-UMU), University Clinical Hospital “Virgen de la Arrixaca”, University of Murcia, El Palmar, 30120 Murcia, Spain; msl70082@um.es; 3Plataforma Sala Blanca, IMIB-Arrixaca, Carretera Madrid-Cartagena S/N, El Palmar, 30120 Murcia, Spain; elena.tristante@imib.es; 4Molecular Recognition and Encapsulation Research Group (REM), Health Sciences Department, Campus de los Jerónimos 135, UCAM-Universidad Católica de Murcia, Guadalupe, 30107 Murcia, Spain; jagabaldon@ucam.edu

**Keywords:** 3D printing, poly(ε-caprolactone), β-tricalcium phosphate, microparticles, composite filament, mesenchymal stem cells, flow cytometry, qRT-PCR, tissue engineering, cell therapies

## Abstract

In this work, we evaluated the influence of a novel hybrid 3D-printed porous composite scaffold based on poly(ε-caprolactone) (PCL) and β-tricalcium phosphate (β-TCP) microparticles in the process of adhesion, proliferation, and osteoblastic differentiation of multipotent adult human bone marrow mesenchymal stem cells (*ah*-BM-MSCs) cultured under basal and osteogenic conditions. The in vitro biological response of *ah*-BM-MSCs seeded on the scaffolds was evaluated in terms of cytotoxicity, adhesion, and proliferation (AlamarBlue Assay^®^) after 1, 3, 7, and 14 days of culture. The osteogenic differentiation was assessed by alkaline phosphatase (ALP) activity, mineralization (Alizarin Red Solution, ARS), expression of surface markers (CD73, CD90, and CD105), and reverse transcription–quantitative polymerase chain reaction (qRT-PCR) after 7 and 14 days of culture. The scaffolds tested were found to be bioactive and biocompatible, as demonstrated by their effects on cytotoxicity (viability) and extracellular matrix production. The mineralization and ALP assays revealed that osteogenic differentiation increased in the presence of PCL/β-TCP scaffolds. The latter was also confirmed by the gene expression levels of the proteins involved in the ossification process. Our results suggest that similar bio-inspired hybrid composite materials would be excellent candidates for osteoinductive and osteogenic medical-grade scaffolds to support cell proliferation and differentiation for tissue engineering, which warrants future in vivo research.

## 1. Introduction

Nowadays, the most common bone regeneration approaches are focused on the fabrication of affordable substitutes to autologous bone grafts [1,2,3,4,5,6]. In this context, a large number of strategies have been successfully developed in recent decades, aiming to produce tailored synthetic soft polymer-based biomaterials (which can be administered via injection), rigid scaffolds that act as a 3D framework to mimic bone structures and functionalities, or a combination of the aforementioned approaches [1,5,7]. An ideal synthetic matrix in bone regeneration and repair needs to be not only bioactive and resorbable, but also must exhibit specific structural characteristics (micro-/macroporosity), mechanical, and biochemical properties to mimic those from native tissues [8,9], since these properties modulate the biological response of the scaffold and also influence its stiffness, surface morphology, hydrophilicity, degradation, cell adhesion, proliferation, and differentiation [10]. As a matter of fact, an interconnected pore network with open porosity facilitates cell penetration and fluid flow, which also enhances the capacity for cell proliferation [7].

To date, around 60% of the bone graft substitutes commercially available are based on ceramics. In this sense, different ceramic precursors have been proposed, but the family of calcium phosphate (CaP)-based ceramics deserves special attention due to their biocompatibility and unlimited availability [11,12,13,14]. Nevertheless, these materials must overcome their main drawback regarding their brittle structure. In addition to ceramic-based substitutes, the most common 3D structured biomaterials are based on natural and synthetic precursors of a different nature, or on their combination [15]. Among the most versatile synthetic polymer precursors for bone graft substitutes, poly(ε-caprolactone) (PCL) deserves special attention due to its biocompatibility, biodegradability, and ductility. Actually, it is widely used for applications in the medical and pharmaceutical industries.

In recent years, a very active branch of materials research based on combining bio-friendly polymers with inorganic ceramics such as β-tricalcium phosphate (β-TCP, β-Ca_3_(PO_4_)_2_) has recently flourished, mainly due to their large potential of applications in clinical orthopedics [13,14,16,17,18,19,20,21]. The association of polymers with TCP not only offers the possibility to fabricate tailored materials with enhanced physicochemical properties, but also with modulable resorption rate, facilitating protein/cell adhesion, proliferation, osteogenic differentiation, and osseointegration [22,23,24,25,26,27,28,29]. These materials could also be used as a carrier for the controlled release of various molecules such as growth factors, antibiotics, bisphosphonates, and statins, which would promote the osteogenesis and regeneration of bone tissue [3,4,5,9,23,30,31,32,33,34,35]. The functionality of the former materials can be exalted by using biological coating additives such as gelatin, chitosan, or fibronectin in order to increase the cell adhesion, avoiding their detachment when they come into contact with organic fluids [36,37,38,39]. In this context, Díaz-Arca et al. [2] reported the fabrication of tricalcium phosphate (TCP) and silicocarnotite (SC) scaffolds via sintering, which mimic the internal microstructure of cancellous bones and can be combined with osteogenic factors to improve their performance for bone reconstruction applications [40]. In such materials, the TCP is responsible for releasing calcium and phosphorus ions, thereby enhancing cell proliferation and cellular differentiation [7]. Similarly, Shin et al. [41] reported the fabrication of biphasic PCL/β-TCP (BCP) composite scaffolds with an interconnected porous structure via salt-leaching and freeze-drying. They reported that such composite materials permit cell survival, accompanied by a significant osteogenic differentiation. However, they were incapable of observing a significant increase in the proliferation of human mesenchymal stem cells (*h*MSCs) beyond seven days in culture. Other interesting findings were reported by Park et al. [42], who demonstrated positive osteogenic differentiation of *h*MSCs under mechanical stimulation of PCL/β-TCP 3D-printed scaffolds. Surprisingly, composites with a lower β-TCP content display lower expression of osteogenic markers if compared to those composites with a higher β-TCP content. However, they only investigated the effects of β-TCP on the proliferation and differentiation in vitro for a nine-day period. More recently, Yang et al. [43] reported the biological responses of MC3T3-E1 cell lines on PCL/β-TCP 3D-printed scaffolds. They demonstrated that those materials treated by oxygen plasma and/or amine plasma-polymerization positively influence the adhesion, proliferation, and osteogenic differentiation of cells. Thus, the fabrication of alternative bioinspired composite material to stimulate bone regeneration and repair still remains an attractive and open issue.

In this work, we focus on the fabrication of novel hybrid polymeric–ceramic porous 3D-printed composite scaffolds based on poly(ε-caprolactone) (PCL) and β-tricalcium phosphate (β-TCP) microparticles, and their influence in the process of adhesion, proliferation, and osteoblastic differentiation of multipotent adult human bone marrow mesenchymal stem cells (*ah*-BM-MSCs) via primary human mesenchymal stem cell culture. Moreover, we assessed the potential positive influence of the release of calcium and phosphorus ions on metabolic activity and cellular differentiation. We also studied the effects of the physico-structural characteristics of the scaffolds on cytotoxicity (cell viability), extracellular matrix production, and variations in gene expression to confirm osteoblastic differentiation.

## 2. Results and Discussion

### 2.1. Study Design

The first step of the present study was the fabrication of poly(ε-caprolactone) (PCL)/β-tricalcium phosphate (PCL/β-TCP) 3D-printed porous disk-shaped scaffolds of 8 × 1.5 mm (diameter × height). Subsequently, a detailed characterization of their structure and properties was performed. Three experimental groups were established as follows: (1) Control group or cells growing on polystyrene (TCPS), (2) plates containing PCL scaffolds, and (3) plates containing PCL/β-TCP scaffolds. Finally, the in vitro biological behavior was investigated using a primary culture of *ah*-BM-MSCs. All biological studies were carried out according to a previously established schedule (Figure 1).

### 2.2. Material Composition and Characterization

#### 2.2.1. Morphological Characterization and Microanalysis of the Composite Filaments and the 3D-Printed Scaffolds

The morphology and diameter of native PCL filaments and PCL/β-TCP composite filaments were examined and characterized before being fed into the 3D bioprinter. As can be seen in the SEM micrographs, the native PCL filaments show a smooth homogeneous surface (Appendix A), while the PCL/β-TCP filaments show a heterogeneous micro-granulated surface due to the presence of β-TCP microparticles (Appendix A). Likewise, the β-TCP coating resulted in a slight increase in the filament diameter from 1.77 to 1.93 mm (Appendix A). Additionally, an EDX analysis was carried out on the coated filaments in order to check the presence of phosphorus and calcium elements, obtaining 33.5% and 66.5% of the total weight, respectively (Appendix A).

Figure 2A–D show typical non-polarized optical microscopy images for native PCL (Figure 2A,B) and PCL/β-TCP (Figure 2C,D) 3D-printed scaffolds. They exhibit a standard disk shape with a diameter of 8 mm and a height of 1.5 mm. As seen in the magnified microscopies, the printed lines display an ordered structure, adopting a parallel distribution in every printed layer (Figure 2B,D). The distance between the scaffold printed lines was 1.30 ± 0.40 mm in the vertical and horizontal layers, and 0.35 ± 0.03 mm in the diagonal layers, providing a network of interconnected pores of triangular morphology with an average porosity of 200 µm. We also clearly observed TCP microparticles randomly distributed throughout the entire PCL/β-TCP scaffold (Figure 2C,D). Finally, the energy dispersive X-ray (SEM-EDX) microanalysis was performed in order to verify the presence of the TCP mineral fraction. According to our analyses, we obtained a share of 32.25% and 67.75% of the total weight of phosphorus and calcium elements, respectively (Figure 2E,F).

#### 2.2.2. Pore Structure Characterization by Micro-CT

Three-dimensional (3D) printing has emerged as a powerful tool for tissue engineering by enabling 3D cell cultures within complex 3D biomimetic architectures [44]. In contrast to the conventional techniques such as electrospinning [45], freeze-drying [46], gas foaming [47], or fiber deposition [48], 3D printing brings more control to both internal and external scaffold geometry. Parameters such as pore size, total porosity, and pore connectivity play an important role in the mass transport of biological fluids, oxygen, nutrients, and cells from the external environment to the inner parts of the scaffold promoting tissue ingrowth [49].

The 3D rendering of the CT images from six randomly selected scaffolds (*n* = 3 per sample) showed no statistically significant differences between the volume of both PCL and PCL/β-TCP scaffolds (Figure 3). Based on the data collected from 3D reconstructions, both types of scaffolds had an average volume (*V_S_*) of 31 ± 1 mm^3^, which corresponds to a global porosity of 0.59 ± 0.10 when compared to the theoretical volume (*V_T_* = 75.4 mm^3^) of an 8 × 1.5 mm (diameter, height) disk.

In order to obtain scaffolds with open and interconnected porosity, we used an 8 mm biopsy punch to homogenize the samples obtaining scaffolds with an absence of a well-defined peripheral limiting layer. This technical detail allows the peripheral pores on its entire surface to be in direct contact with the external environment. In addition, we selected a triangular infill pattern, which results in a network of interconnected pores.

Finally, the 3D rendering of PCL/β-TCP scaffolds (Figure 3B,D,F) showed a random distribution of β-TCP microparticles over the entire scaffold volume, as outlined in the previous section (Figure 2D).

### 2.3. Influence of the Scaffold Composition

#### 2.3.1. Protein Adhesion: Coomassie Brilliant Blue Test

The Coomassie stain is one of the most widely used assays for protein quantification. As a matter of fact, it provides sensitive protein detection, along with simplified protocols, and it is relatively accurate for most proteins. Besides, it is well known that the contact of a living body with a material induces the absorption of a protein monolayer on its surface, depending on the nature of the material, creating an interface where other proteins and cells adsorb [50]. This protein layer promotes the fact that cell adhesion receptors located on the cell membrane (such as integrins) can recognize the arginine–glycine–aspartate (RGD) peptide, creating anchor points on the surface of the biomaterial [51]. In addition, overall porosity, size, pore distribution, and particle size are factors that also influence the degree of protein adsorption. Thus, the existence of pores greatly increases the surface area of the materials and enhances protein adsorption. In this context, Zhu et al. [52] demonstrated that the amount of total proteins adsorbed by porous Ca-P biphasic Ca-P porous ceramic (BCP) (HA/TCP 1⁄4 7:3) was higher than that of dense BCP. Furthermore, once the cells have adhered, the processes of the synthesis and release of the ECM constituent molecules such as osteopontin (OPN), collagen type I (COL1A1), and bone morphogenetic proteins (BMPs), among other regulators of osteogenesis begin [51,53]. In other words, the proteins initially adsorbed on the surface and coming from the fluids provide a temporary substrate for cell adhesion.

In our study, the Coomassie Brilliant Blue test—a simple and reliable qualitative procedure—showed that both PCL and PCL/β-TCP scaffolds were able to absorb proteins on their surface (Figure 4). The scaffolds that were not submerged in fetal bovine serum (FBS) lost their bluish color after three washes with the destaining solution (Figure 4(2)), while those scaffolds immersed in FBS for 30 min remained blue after the washing step (Figure 4(3)). It is also worth noting that apparently, native PCL scaffolds exhibit a slightly light color if compared to PCL/β-TCP after the washing step. The latter is probably due to the presence of β-TCP microparticles, which would have slightly modified the hydrophobic nature of the surface of PCL making them more hydrophilic, facilitating the proteins adhesion.

#### 2.3.2. In Vitro Degradation Kinetics

The degradation kinetics of the polymer used for the fabrication of medical-grade composites are of paramount importance. In fact, it is highly desirable to match the polymer biodegradation with their intended functional tissue regeneration use. Typically, degradation is assessed via scaffold immersion in a buffered solution (such as commercially available DMEM or homemade simulated body fluid (SBF) solutions) during periods ranging from days or weeks to months. The scaffold weight loss is directly correlated with the polymer degradation process. PCL is considered a polymer with a very long degradation time. In our case, we immersed at 37 °C into a biomimetic SBF solution all of our PCL and PCL/β-TCP scaffolds. The immersion time ranged over 3, 7, 14, 21, and 28 days according to different authors [54,55]. The percentage of the scaffold’s weight loss immersed in SBF solutions for various periods of time was then calculated using Equation (2) [56,57,58]. As expected, for the periods studied, the PCL degradation was less than 1%, even for those exposed during 28 days (data not shown). These results are in total agreement with the expected ones for similar PCL-based composite materials. In fact, as reported by Sukanya et al. [54], the PCL is expected to exhibit weight losses of around 2% for an extended period of immersion in SBF of 90 days. In spite of the fact that we did not study the TCP degradation, it is worth mentioning that it is widely known from the literature that β-TCP bioceramics degrade relatively quickly, being able to simultaneously promote new bone formation in vivo [59]. In addition, from a physiological point of view, β-TCP degrades faster than hydroxyapatite, either via chemical dissolution or via reabsorption by phagocytic cellular mechanisms (macrophages or osteoclasts). The latter would directly affect the bone remodeling process [60].

### 2.4. Cell Viability and Proliferation

#### 2.4.1. *ah*-BM-MSC Characterization

In the present study, *ah*-BM-MSCs were used to evaluate the biocompatibility and the osteogenic activity of PCL/β-TCP scaffolds due to their clinical importance and their great capacity for osteoblastic differentiation [61]. Basically, MSCs have the ability to differentiate ex vivo into various cell lines (such as osteoblast) under favorable environmental conditions [62]. As can be seen in Table 1, the high positive expression of mesenchymal stem cell-like markers, specifically CD73, CD90, CD105, and CD44, was observed in more than 97% of cells isolated, manifesting its immature phenotype and important regenerative potential and making them suitable for in vitro assays, as reported previously in the literature [63].

#### 2.4.2. Cytotoxicity Assay

Biocompatibility is one of the most important factors to be taken into account for the fabrication of scaffolds for tissue engineering. In this regard, both PCL and β-TCP have been previously reported as biocompatible and safe biomaterials [64,65,66]. Cytotoxicity assays are widely used to measure loss of cellular or intracellular structures and functions, including lethal cytotoxicity levels, providing an unequivocal indication of their potential to cause cell or tissue damage. The cell viability was assessed by using the AlamarBlue assay after seeding *ah*-BM-MSCs on PCL and PCL/β-TCP scaffolds (Figure 5). The results of cell viability at 24 and 72 h after seeding demonstrated that both the PCL and PCL/β-TCP scaffolds had no cytotoxic effects on *ah*-BM-MSCs. The values of cell viability ranged from 86% to 114% for PCL and PCL/β-TCP scaffolds at 24 h, and from 88% to 115% at 72 h. The mean percentage of viability after direct seeding on both scaffolds was 98 ± 11 (PCL) and 100 ± 10 (PCL/β-TCP) at 24 h, and 101 ± 9 (PCL) and 98 ± 12 (PCL/β-TCP) at 72 h. It is also worth noting that no statistically significant differences were observed between the groups.

#### 2.4.3. Cellular Metabolic Activity Assay

Poly(ε-caprolactone)-based scaffolds have been widely used in recent years for tissue engineering applications due to their biocompatibility, biodegradability, structural stability, and mechanical properties [67]. In this sense, different authors have demonstrated that cells are able to adhere and proliferate onto different PCL-based scaffold structures, such as electrospun fibers [68] and 3D-printed scaffolds [69]. In our work, the hydrophilic properties provided by the coating with the β-TCP microparticles based on a preliminary wettability test using distilled water (data not shown in this work) may favor adhesion, initial cell growth, and differentiation, since both *ah*-BM-MSCs and osteoblasts show more affinity toward hydrophilic surfaces [70]. For this reason, the wettability of the scaffold surface was one of the key criteria we took into account when designing this study.

The cellular metabolic activity of *ah*-BM-MSCs grown on both PCL and PCL/β-TCP scaffolds was evaluated using the AlamarBlue assay on days 1, 3, 7, and 14 after seeding (Figure 6). The viability of cells growing on scaffolds was normalized with respect to cells growing on plastic (TCPs), which were taken as a positive control. The viability increased gradually for the different time periods studied, indicating that the cells successfully adhered to the scaffolds with a good proliferation rate. It is worth noting that the PCL/β-TCP scaffolds showed lower viability than native PCL scaffolds at early periods (days 1 and 3); however, both viability rates were similar at 7 and 14 days of study, either with growth medium (GM) or after the addition of osteogenic medium (OM) on day 7. As clearly shown in the plot, no significant differences were found between scaffolds at these time periods. However, we noted that the addition of OM caused a slight increase in the metabolic activity with respect to those cells seeded on PCL/β-TCP scaffolds with GM at 14 days.

### 2.5. Cell Differentiation Studies

#### 2.5.1. Alkaline Phosphatase (ALP) Activity

ALP is a hydrolase enzyme responsible for dephosphorylating molecules under alkaline conditions (pH > 10). It is present within all tissues of the body, particularly in bone cells, and is considered an early indicator of osteoblastic differentiation [71]. As widely discussed in the literature, an ALP assay allows establishing a direct relationship between the presence of ALP activity and the normal development of bone in the human body [72,73].

The ALP activity of cultured *ah*-BM-MSCs on both PCL and PCL/β-TCP scaffolds is shown in Figure 7. From days 7 to 14 (GM), no significant difference in ALP activity was observed between the different time periods or within scaffold types. On the contrary, ALP activity significantly increased from days 7 to 14 (OM). The maximum ALP activity values were noticed at 14 days in the presence of osteogenic medium (OM) on cells seeded onto PCL/β-TCP scaffolds, which reveals that β-TCP microparticles seem to exert an impact on osteoblastic phenotype and enhance cellular responses. As a matter of fact, previous studies have indicated that the addition of β-TCP granules into PCL-based composites might enhance the osteoblastic differentiation ability of bone mesenchymal stem cells (BMSCs) [74]. Nevertheless, there still exists a current controversy regarding the time period in which the ALP activity shows its maximum activity. In this context, Chen et al. [75] studied the osteoblastic response of MSCs on hyaluronic acid (HA)/β-TCP polymeric-based scaffolds. They reported a well-defined peak in ALP activity by day 7, followed by a decrease, while Jensen et al. [76] reported the maximum ALP activity after 14 days of study in similar polymer-based composites.

#### 2.5.2. In Vitro Mineralization. Alizarin Red Solution (ARS) Staining

Calcium deposition or in vitro mineralization are late markers of osteogenic differentiation. Deposited calcium can be quantified or stained with Alizarin Red Solution (ARS) stain, showing a positive staining (red) of mineralized nodules [77]. The osteogenic effect of the manufactured scaffolds was verified by determining the presence of calcium deposits in the cell culture. For this purpose, the Alizarin Red expression of *ah*-BM-MSCs cultured on PCL and PCL/β-TCP scaffolds was assessed at 7 and 14 days after seeding (Figure 8). No visual differences were observed after examining the stained samples at different time periods under a contrast phase microscope (data not shown). However, the quantitative examination showed that the Alizarin Red activity of *ah*-BM-MSCs increased significantly from days 7 to 14 with both growth medium (GM) and osteogenic medium (OM). The addition of OM promoted a slight increase in the controls at 14 days, but no significant differences were observed between the PCL and PCL/β-TCP scaffolds.

#### 2.5.3. Monitoring Surface Markers on *ah*-BM-MSCs Seeded on the Scaffolds

To date, the most commonly reported positive markers related to adult human MSC (*ah*MSCs) surface are CD105, CD90, CD44, CD73, CD29, CD13, CD34, CD146, CD106, CD54, and CD166 [78]. The minimal criteria of MSCs include: (i) Remaining plastic-adherent under standard culture conditions; (ii) expressing CD73, CD90, and CD105; and (iii) differentiating into osteoblasts, adipocytes, and chondrocytes in vitro [79]. The positive expression of the cluster of differentiation markers CD73, CD90, and CD105 of *ah*-BM-MSCs seeded on both PCL and PCL/β-TCP scaffolds is shown in Figure 9. The results are expressed as the percentage of marker lost after 14 days of culture with both GM and OM (added on day 7). As observed, by day 14, the CD73 and CD90 markers’ expression decreased significantly after the addition of OM. For both markers, the PCL/β-TCP scaffolds showed a higher marker loss than the PCL scaffolds and controls. The higher marker loss throughout the study affected the CD105 marker, whose expression decreased significantly after the addition of OM. In this case, the PCL/β-TCP scaffolds showed similar marker loss than the PCL scaffolds. This decrease in the positive expression of the mesenchymal markers (CD73, CD90, and CD105) could be a consequence of the osteogenic differentiation program activation [80].

#### 2.5.4. Osteogenic Gene Expression: Quantitative Real-Time Polymerase Chain Reaction (qRT-PCR) Assay

The gene expression of osteogenic-specific markers such as ALPL, COL1A1, RUNX2, BGLAP, IBSP, SPARC, and SPP1 was used to determine osteoblastic differentiation in vitro by qRT-PCR (Figure 10). As observed, the ALPL marker displayed an increase in gene expression throughout the study, showing higher values after the addition of osteogenic medium (OM) by day 7. In this time period, the PCL/β-TCP scaffolds exhibited higher ALPL gene expression than the PCL scaffolds. For its part, the COL1A1 marker displayed a significant peak in gene expression at 14 days with growth medium (GM). Likewise, the COL1A1 expression with OM increased for the PCL/β-TCP scaffolds, with no significant differences in the time periods studied. In the same way, the exhibited RUNX2 gene expression remains almost constant from days 7 to 14 with GM. However, a non-negligible decrease in gene expression was observed by day 14 with OM. The BGLAP gene expression also exhibited almost constant values throughout the study. As a matter of fact, no significant differences were observed between the different time periods. In contrast, the IBSP expression showed a prominent peak on both the PCL and PCL/β-TCP scaffolds on day 14 with GM. However, after the addition of OM, an abrupt decrease in IBSP expression was observed. The SPARC gene expression showed higher values in the presence of GM but lower marker expression in the presence of OM. Finally, the SPP1 marker showed a clear decreasing trend from days 7 to 14 with both types of culture media, either GM or OM. It is worth mentioning that in the latter case, the lower expression values were observed on day 14 in the presence of OM.

These results are in total accordance with the results reported by Jensen et al. [76], who studied the influence of PCL scaffolds functionalized with hyaluronic acid and β-TCP on dental pulp stem cells. In this study, alkaline phosphatase, runt-related transcription factor 2, bone sialoprotein, and osteopontin markers showed similar trends to those achieved in our work. Similar results were reported by Rabadan-Ros et al. [81], who evaluated the impact of a porous Si-Ca-P monophasic ceramic on osteogenic differentiation of adult human mesenchymal stem cells (*ah*MSC) in a 28-day study, obtaining an up-regulation of alkaline phosphatase, collagen type I, osteopontin, integrin-binding sialoprotein, and osteonectin expression, while runt-related transcription factor 2 and osteocalcin expression remained relatively unchanged along the assay. Considering the molecules necessary for the mineralization of the extracellular matrix, Runt is defined as a transcription factor in *Drosophila*, which has affinity with the α subunit of Core Binding Factor Alpha (Cbfa). The three related genes called Core Binding Factor Alpha 1 (Cbfa1), Cbfa2, and Cbfa3 are capable of generating different proteins [82,83,84]. Cbfa1 (Runx2) is the specific transcription factor for osteogenesis (Runx2/AML3) and the determinant for the differentiation of mesenchymal cells toward osteoblastic lineage [85] under the stimulus of specific genes such as OC, ALP, BSP, collagen type I (Col 1), and Collagenase-3 (matrix metalloproteinase-13 (MMP-13) [86].

In this sense, the trend observed in RUNX2 expression in our study and in others analogous could be attributed to the fact that it is involved in multiple signal transduction pathways, and its activity is tightly regulated at both the transcriptional and posttranslational levels [87].

## 3. Materials and Methods

### 3.1. Fabrication of Polymer-Based Porous 3D-Printed Scaffolds

#### 3.1.1. Preparation of PCL Filaments Coated with β-TCP Microparticles

The process used for the preparation of the poly(ε-caprolactone)/β-TCP composite filaments, labeled as PCL/β-TCP, is described herein. Briefly, PCL (MW 50 kD) filaments of 1.75 mm in diameter and 150 mm in length were heated at 65 °C in hot water (bain-Marie) until the filament became flexible and changed its aspect from opaque white to completely transparent. After this, the filament was coated manually bearing on a surface dusted with copious amounts of β-TCP powder previously mortared and sieved at 125 µm. The collected coated filaments were dried at room temperature (RT) for 30 min before being fed into the printer extruder. It is worth mentioning that both the PCL (uncoated) and PCL/β-TCP filaments were weighed in order to estimate the final β-TCP microparticle concentration of 5 wt%. The average filament diameter was calculated from at least 20 measurements performed on different optical images acquired with a stereomicroscope.

#### 3.1.2. Design and Printing of 3D Scaffolds with Controlled Porosity Using the Fused Deposition Modeling (FDM) Method

The scaffolds were designed with the software REGEMAT 3D Designer v1.4.4 and manufactured by the fused deposition modeling (FDM-3D) method using a REGEMAT 3D Bio V1^®^ bioprinter (REG4Life, REGEMAT 3D, Granada, Spain) equipped with a glass bed and a 0.4 mm diameter nozzle (Figure 11A). Both the PCL and PCL/β-TCP scaffolds dimensions were set to 1.50 × 20 × 20 mm (height × width × length) (Figure 11B,C) and printed using the same parameters (infill speed of 11 mm/s, layer height of 0.25 mm, pore size of 200 µm, and printing temperature of 160 °C). The number of perimeters and solid bottom/top layers was set to 0, allowing to obtain scaffolds with interconnected and open porosity. In order to enhance the reproducibility of the experiments and to reduce the variability between the samples, an 8 mm biopsy punch was used to prepare defined and reproducible scaffolds (Figure 11D), obtaining four disk-shaped scaffolds of 8 × 1.5 mm (diameter × height) per printed scaffold (Figure 11E), which complies with the recognized international standard ISO/FDIS 23317 [88].

Then, the scaffolds were rinsed with distilled water in an orbital shaker for 30 min at 250 rpm to remove any undesired dust contamination before being sterilized with a pulsed light system XeMaticA-Basic-1L (Steribeam, Kehl, Germany). The latter produces pulses ranging from infrared (IR) to ultraviolet (UV) light with 21% of UV content [89]. The effectiveness of the method was verified by incubating the samples for five days at 37 °C in test tubes containing 7 mL of sterile Tryptic Soy Broth (#T8907; Sigma-Aldrich, Saint Louis, MO, USA). This methodology is widely used for detecting the presence of microorganisms.

### 3.2. Characterization of the Composite Filaments and the 3D-Printed Scaffolds

#### 3.2.1. Morphological Characterization and Microanalysis of the Filaments and the 3D Scaffolds

##### Scanning Electron Microscopy and Energy-Dispersive X-ray Analysis (SEM-EDX)

Morphological characterization of the filaments and the 3D-printed scaffolds was carried out by means of a digital camera (Axiocam 305 color) coupled to an optical microscope (Zeiss 415510), scanning electron microscopy (SEM; model JEOL-6100 (JEOL Ltd., Tokyo, Japan)), and energy dispersive X-ray spectroscopy (EDX; Oxford INCA (Oxford Instruments plc., Abington, Oxfordshire, U.K.). For SEM analysis, the samples were coated with gold, while for EDX analysis, they were carbon-coated to avoid spectrum overlaps with other elements.

##### Micro-Computed Tomography (µCT) Scaffold Imaging: Porosity and β-TCP Particle Distribution

The scaffold porosity and the distribution of the β-TCP particles in the scaffold were detected using a Quantum GX2 micro-CT imaging system (PerkinElmer, Hopkinton, MA, USA) at a voxel size of 72 µm. Typically, three samples randomly selected were placed on a 35 mm diameter object bed. A complete scan (scanning parameters: energy = 90 kV; intensity = 88 µA) was then performed. The scanned microstructural images were reconstructed using Invesalius 3.1 software (^©^2007–2017, Center for Information Technology Renato Archer CTI). The total porosity, pore size, and open porosity of each scaffold were analyzed. In order to obtain the total porosity *p*, the scaffold volume V_S_ was compared to the theoretical volume of the cylinder V_T_ using Equation (1) [10]:*p* = 1 − (V_S_/V_T_)(1)

#### 3.2.2. Protein Adhesion: Coomassie Brilliant Blue Test

The Coomassie Brilliant Blue test was used to determine protein adsorption on the surface of both the PCL and PCL/β-TCP scaffolds. First, the scaffolds were immersed for 30 min in fetal bovine serum (FBS; #F7524, Sigma-Aldrich, Saint Louis, MO, USA) to allow protein adhesion. Then, the excess solution was removed and the scaffolds were left to dry in an oven at 37 °C for 45 min. After drying, the scaffolds were fixed with 4% paraformaldehyde in PBS for 15 min and immersed for 30 min in the Coomassie staining solution, which was prepared according to the manufacturer’s instructions (Coomassie Brilliant Blue R-250 Dye, #20278; Thermo Fisher, Rockford, IL, USA). Finally, the scaffolds were rinsed three times with a destaining solution containing methanol/acetic acid/distilled water (40/10/50, % *v*/*v*/*v*).

#### 3.2.3. In Vitro Degradation Kinetics

Scaffold degradation was assessed in vitro by measuring the weight loss of scaffolds. Typically, polymeric scaffolds were weighed and subsequently immersed in 2 mL of complete GM (DMEM) in 12-well plates, incubated at 37 °C, in a 5% CO_2_ atmosphere with 95% relative humidity, for different time periods of 1, 3, 7, 14, 21, and 28 days. Then, scaffolds were recovered from the medium, rinsed with deionized water, dried at 37 °C overnight, and re-weighed. The weight loss (WL) was calculated by applying the following equation:WL% = [(W_0_ − W_d_)/W_0_] × 100(2)
where W_0_ and W_d_ indicate the weight of the scaffold before and after the scheduled immersion time, respectively. Degradation experiments were performed, at least in triplicate, and each measurement was performed six times to obtain appropriate statistics.

### 3.3. Isolation, Characterization and Culture of Adult Human Bone Marrow-Derived Mesenchymal Stem Cells (ah-BM-MSCs)

The *ah*-BM-MSCs were isolated and cultured as previously described [90,91] and characterized in accordance with the criteria established by the International Society for Cell Therapy (ISCT) [92] (data not shown). Briefly, three healthy patients scheduled for elective orthopedic surgery were recruited for this study. Informed consent was obtained from each of them. Bone marrow was collected by percutaneous direct aspiration from the iliac crest. The mononuclear cells were then separated from the bone marrow and washed using a SEPAX^®^ S-100 device (Biosafe, Eysins, Switzerland). For more details related to the methodology and procedure applied, cell isolation, culture, and expansion of *ah*-BM-MSCs, please refer to previous publications [90,91]. After determining nucleated cells viability with trypan blue solution (#T8154; Sigma-Aldrich, Saint Louis, MO, USA) for initial expansion, the harvested cells were seeded at a density of 3.75 × 10^5^ mL in a 75 cm^2^ tissue culture flask (Biofil^®^) with 10 mL of basic growth culture medium (GM) consisting of Dulbecco’s Minimal Essential Medium (DMEM) (#31885-023; Gibco, Bleiswijk, the Netherlands) incorporating 10% (*v*/*v*) inactivated fetal bovine serum (FBS) (#F7524, Sigma-Aldrich, Saint Louis, MO, USA), 100 U/mL of penicillin, and 100 μg/mL of streptomycin (#P4333, Sigma-Aldrich, Saint Louis, MO, USA), and then incubated at 37 °C in a 5% CO_2_ atmosphere with 95% of relative humidity. After seven days, the culture medium was renewed, thus eliminating the non-adherent cells, including some hematopoietic cells, facilitating the identification and selection of the attached cells [93]. When 80–90% confluence was reached, cells were treated with 0.25% *w*/*v* trypsin/EDTA (#T4049, Sigma-Aldrich, Saint Louis, MO, USA) in phosphate-buffered saline (PBS, pH 7.4) for 5 min. The collected cells were then subcultured at a 1:3 ratio and expanded for future use. Passages 3 and 4 (P3-P4) were trypsinized and collected to be used in all subsequent in vitro assays.

All biological experiments were in full compliance with regulatory guidelines, and the experimental protocol was ethically reviewed and approved by the Institutional Ethics Committee of UCAM-Universidad Católica de Murcia (Authorized No CE051904) UCAM ethics committee (CE nº 052114).

#### 3.3.1. *ah*-BM-MSC Characterization

Before performing the in vitro assays, the purity of the *ah*-BM-MSCs populations was assessed by flow cytometry (Beckman Dickinson & Co., Franklin Lakes, NJ, USA; Software Navios). In order to characterize the cells, a pool of *ah*-BM-MSCs detached from different flasks was labeled with an MSC Phenotyping Kit (#130-095-198, Miltenyi Biotec, Bergisch-Gladbach, Germany) and PE Anti-Rat CD44H mouse IgG2A antibody (R&D Systems), in order to quantify the expression of CD73, CD90, CD105, and CD44 markers.

#### 3.3.2. Cell Seeding Methods

Prior to cell seeding, the scaffolds were conditioned with the culture medium and incubated at 37 °C in a 5% CO_2_ atmosphere with 95% of relative humidity for 48 h.

For cell adhesion, proliferation, and viability assays, the *ah*-BM-MSCs were seeded onto the top of the PCL and PCL/β-TCP scaffolds at a density of 5 × 10^4^ cells cm^−2^ in 48-well plates, taking as a positive control cells seeded onto tissue culture-treated polystyrene wells (TCPS; Sigma-Aldrich, Corning, NY, USA). For the cellular metabolic activity assay (AlamarBlue^®^ Assay), the scaffolds were rigorously changed to a new 48 well-plate 24 h after seeding, in order to quantify solely the metabolic activity of the cells growing on the scaffolds. At this time, the cells adhered to the bottom of the polystyrene wells were counted and discarded. The mean total count of cells adhered to the bottom of the well plate was 2 × 10^4^ cells cm^−2^, giving an approximate value of 3 × 10^4^ cells adhered to the scaffold.

To perform osteogenic differentiation studies, the scaffolds were placed in 0.4 μm pore culture well inserts (Falcon^®^) and the *ah*-BM-MSCs were seeded at the bottom of the wells at a density of 5 × 10^3^ cells cm^−2^. Cells seeded onto tissue culture-treated polystyrene (TCP) wells were taken as a positive control. From day 7, a set of at least three plates initially cultured in growth medium (GM) were replaced by osteogenic differentiation medium (OM) (OsteoMAX-XF^™^ Differentiation Medium; #SCM121, Sigma-Aldrich, Saint Louis, MO, USA). Thus, the cellular differentiation induced by all of the scaffolds researched in the present study growing either on GM and OM were characterized in order to assess their effectiveness. The media were changed twice a week for all of the experiments performed.

### 3.4. Cell Viability, Adhesion, and Proliferation Assays

#### 3.4.1. Cytotoxicity Assay

The viability of *ah*-BM-MSCs was evaluated using a resazurin-based cell viability assay (AlamarBlue^®^; #DAL1100, Invitrogen, Carlsbad, CA, USA) on days 1 and 3 after seeding. Briefly, at different time periods of the study, fresh medium (500 μL) containing 10% (*v*/*v*) AlamarBlue^®^ reagent was added to each well and incubated at 37 °C in a 5% CO_2_ atmosphere with 95% of relative humidity for 4 h. During this time, the culture plate was wrapped with aluminum foil in order to provide a dark environment. Then, 150 µL aliquots of each well were transferred to a black-walled 96-well plate and fluorescence was measured with a Synergy MX ultraviolet-visible (UV-Vis) spectrophotometer (Bio Tek Instruments Inc., Winooski, VT, USA) at excitation and emission wavelengths of 530 and 590 nm, respectively.

#### 3.4.2. Cellular Metabolic Activity Assay

AlamarBlue^®^ was used according to the manufacturer’s instructions to assess the metabolic activity of the *ah*-BM-MSCs on days 1, 3, 7, and 14 after seeding. In this assay, we only evaluated the metabolic activity of the cells adhered to the scaffolds (cells growing on the well plate are counted and discarded). At each time period of the study, the fluorescence was measured with a Synergy MX ultraviolet-visible (UV-Vis) spectrophotometer (Bio Tek Instruments Inc., Winooski, VT, USA) at excitation and emission wavelengths of 530 and 590 nm, respectively.

### 3.5. Osteoblastic Differentiation Assays

#### 3.5.1. Alkaline Phosphatase (ALP) Activity

The ALP activity of the *ah*-BM-MSCs was assessed at 7 and 14 days after seeding using an Alkaline Phosphatase Detection Kit (#SCR004, Merck Millipore, Billerica, MA, USA). At each time period of the study, cells were detached and an aliquot of 2 × 10^4^ cells (per sample) was treated following the manufacturer’s instructions in order to quantify the hydrolysis of p-nitrophenyl phosphate into phosphate and p-nitrophenol. The reaction affords a yellow-colored by-product that is proportional to the amount of ALP present within the reaction. At each time period of the study, the absorbance was measured at a wavelength of 405 nm in a Synergy MX ultraviolet-visible (UV-Vis) spectrophotometer (Bio Tek Instruments Inc., Winooski, VT, USA).

#### 3.5.2. In Vitro Mineralization of Alizarin Red Solution (ARS) Staining

The in vitro mineralization was evaluated by the specific binding of Alizarin Red Solution (ARS) staining to calcium deposits at 7 and 14 days after seeding using an Osteogenesis Assay Kit (#ECM815; Millipore, Billerica, MA, USA). Briefly, at each time period of the study, the *ah*-BM-MSCs were stained with ARS and visualized using an optical microscope (Motic AE2000, Shimadzu Corp., Kyoto, Japan). Then, to quantify matrix mineralization, the samples were treated following the manufacturer’s instructions, and the optical density (OD) at 405 nm was measured in a Synergy MX ultraviolet-visible (UV-Vis) spectrophotometer (Bio Tek Instruments Inc., Winooski, VT, USA).

#### 3.5.3. Monitoring Surface Markers in the Cells Seeded on the Scaffolds: Cluster of Differentiation (CD)

In order to characterize the *ah*-BM-MSCs, quantification of cell surface markers CD73, CD90, CD105 was performed by flow cytometry (Beckman Dickinson & Co., Franklin Lakes, NJ, USA; Software Navios). First, the cells were detached with TrypLE^™^ Select Enzyme (1X) (#11598846, Gibco, Bleiswijk, the Netherlands) and collected, using 2 × 10^5^ cells for each experimental condition. Then, the cells were washed with PBS and labeled with an MSC Phenotyping Kit (#130-095-198, Miltenyi Biotec, Bergisch-Gladbach, Germany), consisting of the following fluorochromes: Allophycocyanin (APC) conjugated with CD73, fluorescein isothiocyanate (FITC) conjugated to CD90, phycoerythrin (PE) conjugated to CD105, and peridinin-chlorophyll cy5.5 conjugated with CD14/CD20/CD34/CD45. Additionally, a test tube containing cells labeled with the cocktail isotype was used as a control. To check cell viability, 100 µL of the binding buffer (1:10 in H_2_O), 5 µL of Annexin V antibody, and 5 µL of propidium iodide (IP) were added to a tube labeled as Annexin V (Immuno Step).

All of the test tubes were kept at 4 °C (in ice) for 15 min in a dark environment, and washed with 1000 µL of PBS EDTA, except for the Annexin V tube, which was washed with a binding buffer. Then, the tubes were centrifuged (300× *g*, 10 min, 4 °C) and the supernatant was discarded. Finally, the labeled pellet was resuspended with 500 µL of PBS/EDTA and the samples were analyzed by flow cytometry.

#### 3.5.4. Osteogenic Gene Expression: Quantitative Real-Time Polymerase Chain Reaction (qRT-PCR) Assay

qRT-PCR assay was performed to analyze the expression of alkaline phosphatase (ALPL), collagen type I (COL1A1), runt-related transcription factor 2 (RUNX2), osteocalcin (BGLAP), integrin-binding sialoprotein (IBSP), osteonectin (SPARC), and osteopontin (SPP1). The total RNA was extracted from cells using an RNAqueous Micro Kit (Invitrogen by Thermo Fisher Scientific, Waltham, MA, USA) according to the manufacturer’s instructions, followed by reverse transcription of mRNA with an iScript cDNA Synthesis Kit (Bio-Rad). The quantitative PCR was performed using SYBR Premix ExTaq (Takara) in QuantStudio 5 (Applied Biosystems). Specific primers for mRNA were purchased from Qiagen (QuantiTech Primer Assays, Hilden, Germany). All measurements were carried out at least in triplicate. The Ct values were converted to relative quantification using the 2ΔCt method by normalizing to glyceraldehyde-3-phosphate dehydrogenase (GAPDH) and hypoxanthine phosphoribosyltransferase 1 (HPRT1) (Qiagen, Hilden, Germany).

### 3.6. Statistic

All data are represented as the mean ± SD. The statistical significance was determined by a two-way ANOVA using GraphPrism 9.0.1 (GraphPad Software Inc., San Diego, CA, USA) for Windows. Comparisons between groups were evaluated with *t*-tests, with the significance level being *p* < 0.05.

## 4. Conclusions

To conclude, in the present work, we presented an innovative strategy of combining polymers and biofriendly ceramics as a platform for bone tissue engineering biomaterials design. We introduced a novel route to fabricate hybrid 3D-printed porous composite scaffolds with open and interconnected porosity based on poly(ε-caprolactone) (PCL) and β-tricalcium phosphate (β-TCP) microparticles. We studied in detail their influence in the process of adhesion, proliferation, and osteoblastic differentiation of multipotent adult human bone marrow mesenchymal stem cells (*ah*-BM-MSCs). We demonstrated their biological response, bioactivity, and biocompatibility via primary mesenchymal stem cell cultures by studying their effects on cytotoxicity (viability) and extracellular matrix production. The mineralization and alkaline phosphatase (ALP) assays revealed that osteogenic differentiation of the *ah*-BM-MSCs increased in the presence of the 3D-printed PCL/β-TCP scaffolds if compared to both the control group and the native PCL scaffolds, which demonstrates the effective interactions between the β-TCP microparticles and cells. The latter was also confirmed by quantifying the percentage of mesenchymal marker loss (flow cytometry) and monitoring the gene expression levels (qRT-PCR) of most of the proteins involved in the ossification process. The calcium ions released are probably the main responsibility of metabolic activity and cellular differentiation. Our findings suggest that similar bio-inspired hybrid composite materials would be excellent candidates for osteoinductive and osteogenic medical-grade biomaterials.

## Figures and Tables

**Figure 1 ijms-22-11216-f001:**
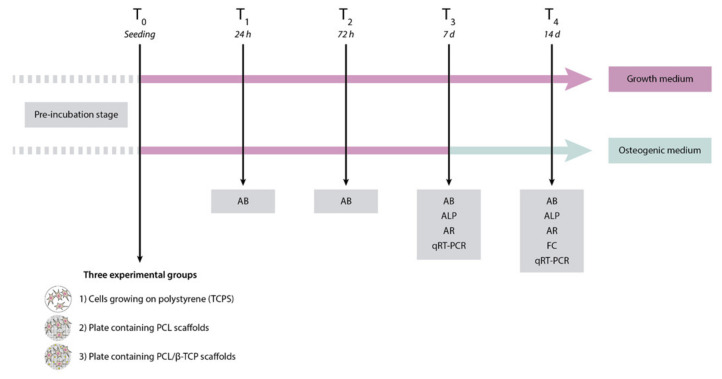
Schematic representation of the study design, including the time schedule of the performed experiments. AB, AlamarBlue assay; ALP, alkaline phosphatase; AR, alizarin red staining; FC, flow cytometry; qRT-PCR, quantitative polymerase chain reaction.

**Figure 2 ijms-22-11216-f002:**
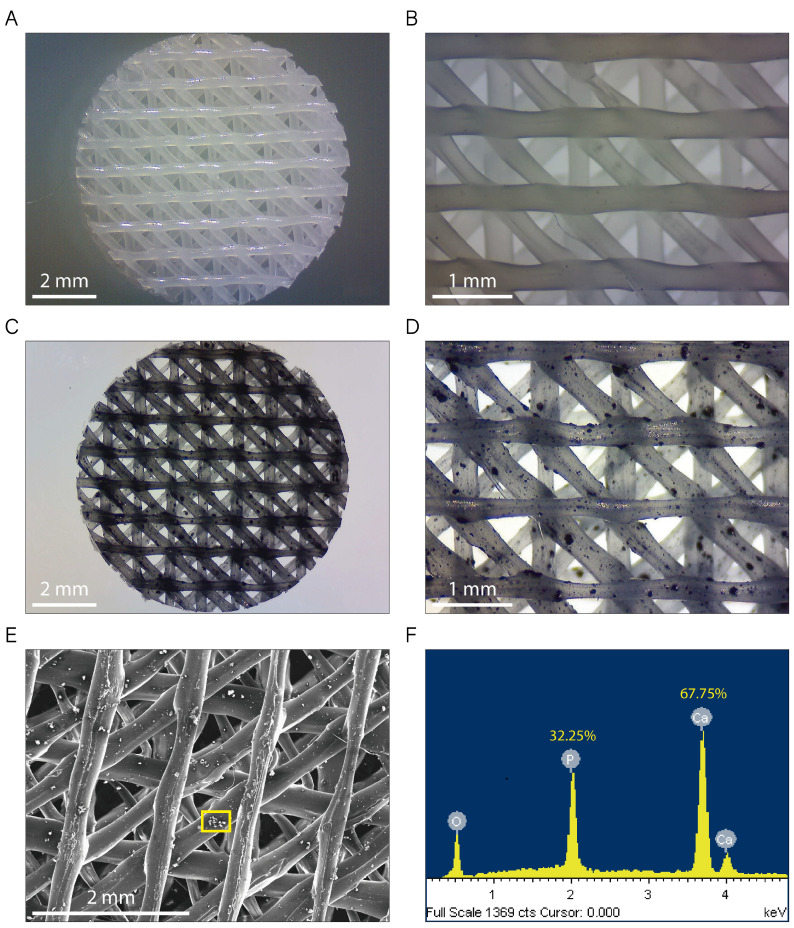
Images of both (**A**,**B**) PCL and (**C**,**D**) PCL/β-TCP scaffolds taken with a stereomicroscope at different magnifications. The β-TCP microparticles were stained with nigrosine for clarity. (**E**) Micrograph acquired with SEM showing a magnification of the PCL/β-TCP scaffold and (**F**) its corresponding EDX spectrum. The researched area is depicted as a yellow square.

**Figure 3 ijms-22-11216-f003:**
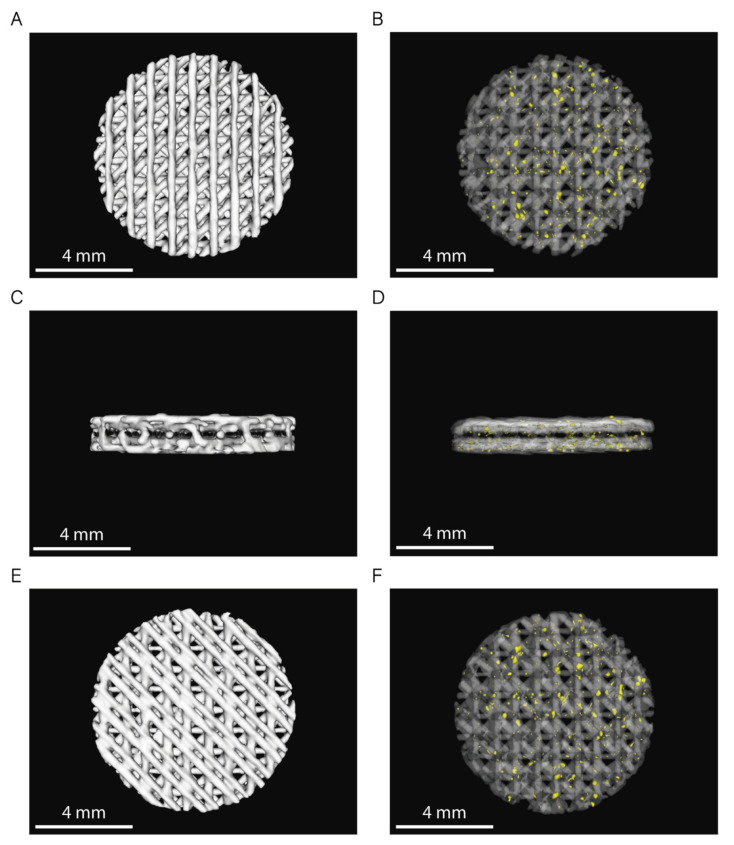
3D rendering of PCL and PCL/β-TCP scaffolds (top: (**A**,**B**); lateral: (**C**,**D**); and bottom: (**E**,**F**) view). In images (**B**,**D**,**F**), PCL is shown with 80% transparency to allow the observation of the β-TCP particles, which are depicted in yellow.

**Figure 4 ijms-22-11216-f004:**
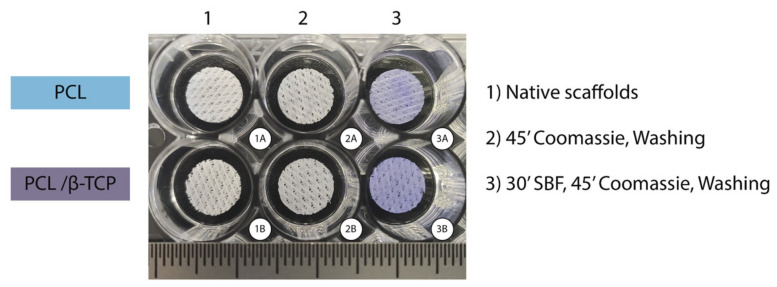
Coomassie Brilliant Blue test performed on PCL (upper row) and PCL/β-TCP (bottom row) scaffolds after non-immersion in FBS (2A,B) and 30 min immersion in FBS (3A,B). Two unstained scaffolds were taken as the control (1A,B).

**Figure 5 ijms-22-11216-f005:**
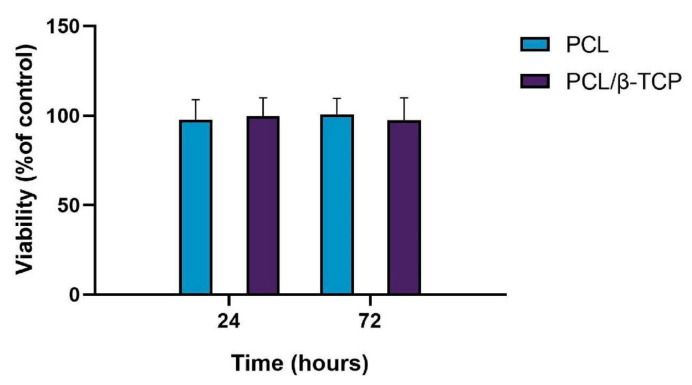
Cell viability using the AlamarBlue assay after 24 and 72 h of direct seeding of *ah*-BM-MSCs on PCL and PCL/β-TCP scaffolds. The mean percentage of viability was calculated and normalized with respect to the viability of cells growing on bare plastic (TCPs) (positive control). Bars represent standard deviations of the mean.

**Figure 6 ijms-22-11216-f006:**
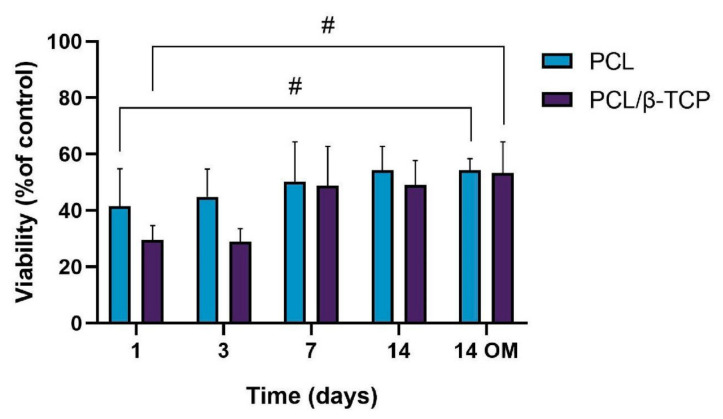
Cellular metabolic activity using the AlamarBlue assay on *ah*-BM-MSCs seeded on both PCL and PCL/β-TCP scaffolds at different time periods. The mean percentage viability was calculated and normalized to the viability of cells growing on plastic (TCPs) (positive control). Bars represent standard deviations of the mean. ^#^ Significant differences between the bracketed groups at different time periods.

**Figure 7 ijms-22-11216-f007:**
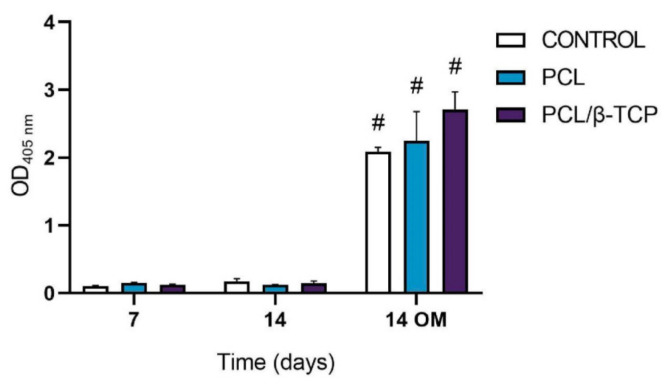
Alkaline phosphatase activity of *ah*-BM-MSCs seeded on PCL and PCL/β-TCP scaffolds after 7 and 14 days of culture. Results are shown as a function of optical density (OD) units. No significant differences were found at the same time periods between the groups. Bars represent standard deviations of the mean. ^#^ Significant differences between the marked group at different time periods (*p* > 0.05).

**Figure 8 ijms-22-11216-f008:**
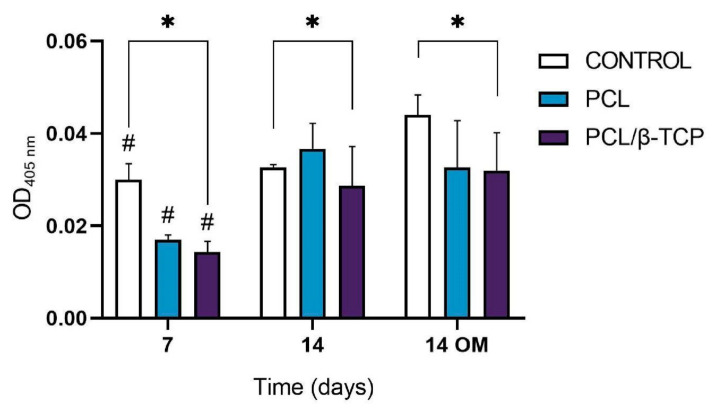
Alizarin Red quantification assay of *ah*-BM-MSCs seeded on PCL and PCL/β-TCP scaffolds after 7 and 14 days of culture. Cells seeded on plastic (TCPs) were taken as the positive control. Results are shown as a function of optical density (OD) units. ^#^ Significant differences between the marked group at different time periods (*p* > 0.05). * Significant differences between the bracketed groups at the same time period.

**Figure 9 ijms-22-11216-f009:**
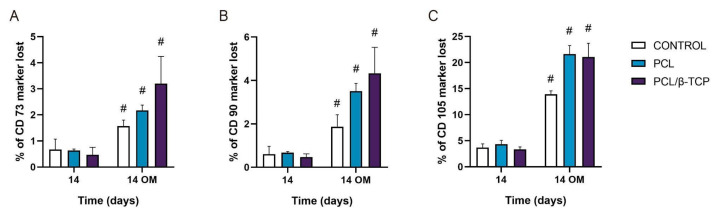
(**A**) CD73, (**B**) CD90, and (**C**) CD105 expressions of the cells seeded on PCL and PCL/β-TCP scaffolds. Data represent the percentage of marker loss at 14 days with different culture medium (GM and OM). The experiment was performed in triplicate. ^#^ Significant differences between the marked group with different culture media.

**Figure 10 ijms-22-11216-f010:**
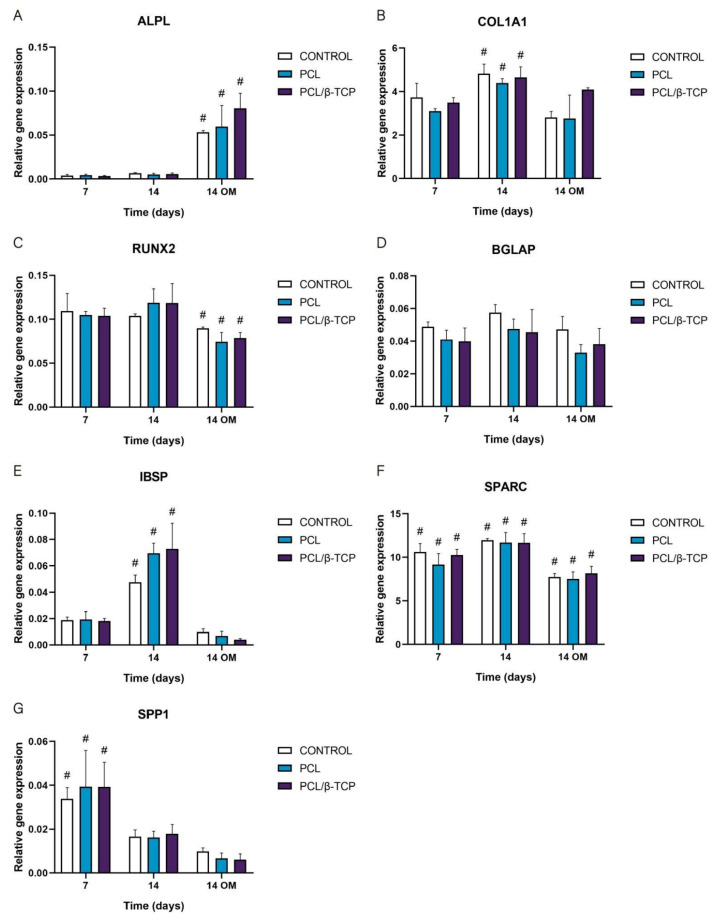
Quantitative real-time polymerase chain reaction results of osteogenic markers after 7 and 14 days of culture. The results are expressed as mean ± SD. ^#^ Significant differences between the marked group at different time periods (*p* > 0.05). No significant differences were observed between the groups in the same time period. (**A**) ALPL, alkaline phosphatase; (**B**) COL1A1, collagen type I; (**C**) RUNX2, runt-related transcription factor 2; (**D**) BGLAP, osteocalcin; (**E**) IBSP, integrin-binding sialoprotein; (**F**) SPARC, osteonectin; (**G**) SPP1, osteopontin.

**Figure 11 ijms-22-11216-f011:**
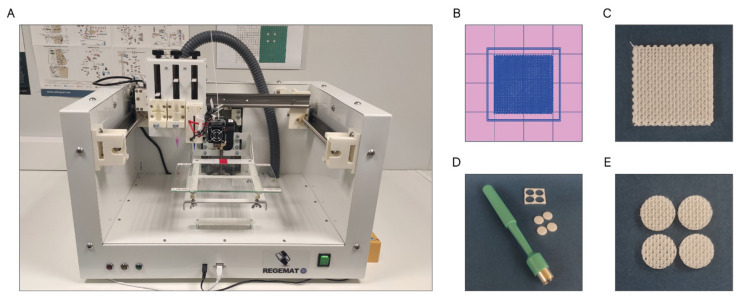
(**A**) REGEMAT 3D Bio V1^®^ bioprinter, (**B**) scaffold design with REGEMAT 3D Designer v1.4.4, (**C**) 3D-printed scaffolds of 1.50 × 20 × 20 mm (height × width × length), (**D**) disk obtention using a 8 mm biopsy punch, and (**E**) disk-shaped scaffolds of 8 × 1.5 mm (diameter × height).

**Table 1 ijms-22-11216-t001:** Flow cytometric analysis for cluster of differentiation (CD) marker expression of *ah*-BM-MSCs at passages 3–4. * Hematopoietic markers.

Antigen	Percentage of Positive Cells
CD73	99.67 ± 0.06
CD90	98.27 ± 0.25
CD105	97.93 ± 0.06
CD44	97.87 ± 0.23
CD14/19/34/45 *	3.10 ± 0.52

## Data Availability

The data presented in this study have been disclosed in the main text.

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
