# Peer review of "Modifications in Gene Expression in the Process of Osteoblastic Differentiation of Multipotent Bone Marrow-Derived Human Mesenchymal Stem Cells Induced by a Novel Osteoinductive Porous Medical-Grade 3D-Printed Poly(ε-caprolactone)/β-tricalcium Phosphate Composite"

_ijms, 2021, doi:10.3390/ijms222011216_

Round 1
Reviewer 1 Report
The manuscript titled "Modifications in gene expression in the process of osteoblastic differentiation of multipotent bone marrow-derived human mesenchymal stem cells induced by a novel osteoinductive porous medical-grade 3D-printed poly(ε-caprolactone)/ β-tricalcium phosphate composite" evaluated the performance of a novel material for culturing adult human bone marrow mesenchymal stem cells. I think the manuscript is very well written overall, very straightforward to follow and understand.
The only suggestion I have is to replace decimal points from “,” to “.”. Other than that, I think this manuscript is ready for publication.
Reviewer 2 Report
The introduction chapter should be shortened as it contains many common phrases.
Figures 5 and 6 are not demonstrative, so they must be removed from the manuscript ( P 8-9). I think that the authors should provide data on the distribution of cells by populations instead of the figure 5.
Cytotoxicity values should be reported as value ± standard deviation (L 268-270)
The authors do not shown photographs of cells on the surface of materials in manuscript.
Reviewer 3 Report
Ivan et al. here developed a PCL-β-TCP scaffold for bone tissue engineering, where they observed upregulation of osteogenesis of MSCs. Unfortunately, their work failed to provide adequate novelty to be published on IJMS. Also, additional experiments should be considered if authors decide to publish somewhere else.
- PCL-β-TCP has been well established in the field of bone tissue engineering. The 3D printing approach has also been demonstrated previously (scientific reports, 2020, 10, 4979; J Mater Res, 2018, 33, 1948; Biomed Res Int, 2018, 2876135; Mater Sci Eng C 2021, 127, 112197; IJMS, 2021, 22, 5409). Therefore, it did not provide any new significance in the field.
- Figures 1 -4 should be organized as one figure, and also, some of the images are not significant enough to be included in the main figure.
- Figure 5 should be in the supporting document, given that it did not provide any useful information.
- Cell attachment image and analysis should be included to showcase that the incorporation of β-TCP can promote cell attachment.
- For cytotoxicity assay, the authors used the AlamarBlue assay, which measured cellular metabolic activities. Often live/dead staining and quantitative analysis should be included to demonstrate the cell viability. Also, the authors should showcase the reduction rate of all the conditions and positive control instead of converting it into % of control.
- Cellular proliferation should also include DNA content assay.
- ALP should be analyzed as ALP/DNA
- Pictures of Alizarin Red S staining should be included, showcasing the mineral deposition.
- Bone specific ECM staining should be performed, including Col 1 and Osteocalcin
- PRC should also analyze the Osteocalcin gene. Also, Day 1 gene expression should be provided as a baseline.
- Figure 12 should be considered as Figure 1 to help the readers understand the experimental conditions.
Round 2
Reviewer 3 Report
Despite authors response, I still think this study lack of novelty to be considered for the journal, due to the lack of novelty from materials and technology aspects. The use of different cell type, and optimal of printing parameter did not guarantee the novelty. Regarding the some specific answers:
- Metabolic activity is inaccurate to reflect the cellular proliferation. ( https://doi.org/10.1111/j.1582-4934.2010.01013.x) and the fact other studies used this is unnecessary to prove it is right. DNA content assay should be included.
- ARS staining is showcasing the distribution of the mineral deposition, uniformed or localized. So it is important.
- PCR expression only indicate gene expression, unnecessarily directly translate into the dECM synthesis. More importantly, PCR is only a snapshot of the gene expression for that particular time point, and the osteogenesis is a continuous process. Col 1 and osteocalcin staining is a must.
- Gene expression should include the start phase, pre-differentiation phase, differentiation phase/mature phase. Day 1 value is important for reader to have a better understanding.
- ALP/DNA is important to provide a better picture, given that difference in cellular proliferation or seeding density can all impact on over ALP reading. ALP/DNA can be more accurate to reflect the cellular osteogenesis.